# Presenilin-1 Familial Alzheimer Mutations Impair γ-Secretase Cleavage of APP Through Stabilized Enzyme–Substrate Complex Formation

**DOI:** 10.3390/biom15070955

**Published:** 2025-07-01

**Authors:** Sujan Devkota, Masato Maesako, Michael S. Wolfe

**Affiliations:** 1Department of Medicinal Chemistry, University of Kansas, Lawrence, KS 66045, USA; sdevkota@ku.edu; 2Mass General Institute for Neurodegenerative Disease, Massachusetts General Hospital, Harvard Medical School, Boston, MA 02131, USA

**Keywords:** protease, amyloid β-peptide, mass spectrometry, fluorescence microscopy

## Abstract

Familial Alzheimer’s disease (FAD) is caused by dominant missense mutations in amyloid precursor protein (APP) and presenilin-1 (PSEN1), the catalytic component of γ-secretase that generates amyloid β-peptides (Aβ) from the APP C-terminal fragment C99. While most FAD mutations increase the ratio of aggregation-prone Aβ42 relative to Aβ40, consistent with the amyloid hypothesis of Alzheimer pathogenesis, some mutations do not increase this ratio. The γ-secretase complex produces amyloid β-peptide (Aβ) through processive cleavage along two pathways: C99 → Aβ49 → Aβ46 → Aβ43 → Aβ40 and C99 → Aβ48 → Aβ45 → Aβ42 → Aβ38. Understanding how FAD mutations affect the multistep γ-secretase cleavage process is critical for elucidating disease pathogenesis. In a recent study, we discovered that FAD mutations lead to stalled γ-secretase/substrate complexes that trigger synaptic loss independently of Aβ production. Here, we further investigate this “stalled complex” hypothesis, focusing on five additional PSEN1 FAD mutations (M84V, C92S, Y115H, T116I, and M139V). A comprehensive biochemical analysis revealed that all five mutations led to substantially reduced initial proteolysis of C99 to Aβ49 or Aβ48 as well as deficiencies in one or more subsequent trimming steps. Results from fluorescence lifetime imaging microscopy support increased stabilization of enzyme–substrate complexes by all five FAD mutations. These findings provide further support for the stalled complex hypothesis, highlighting that FAD mutations impair γ-secretase function by promoting the accumulation of stalled enzyme–substrate complexes.

## 1. Introduction

Alzheimer’s disease (AD), a form of dementia characterized pathologically by cerebral deposits of 42-residue amyloid β-peptides (Aβ42), is the most prevalent neurodegenerative disorder, primarily associated with aging, affecting 5% of people aged 65 to 74, 13% of people aged 75–84, and 33% of people aged 85 and older [1]. While most cases of AD are sporadic and late-onset, a small subset (1–5%) suffer from familial Alzheimer’s disease (FAD), which is dominantly inherited and typically manifests in midlife. FAD is caused by missense mutations in genes encoding the amyloid precursor protein (APP) or presenilin [2,3,4], the catalytic component of the γ-secretase complex [5] that produces Aβ from transmembrane proteolysis of APP C-terminal substrate C99. Over 200 FAD-associated mutations have been identified in the two presenilin genes (presenilin-1 and -2; *PSEN1* and *PSEN2*), with >90% of these in PSEN1. Understanding the biochemical consequences of FAD mutations on γ-secretase proteolytic processing of C99 to Aβ can provide critical clues to the molecular underpinnings of AD pathogenesis.

γ-Secretase is a multi-subunit, membrane-embedded aspartyl protease complex composed of presenilin, nicastrin, Aph1, and Pen-2 [6] that catalyzes the intramembrane cleavage of ~150 type I transmembrane proteins, including APP, after ectodomain shedding by membrane-tethered proteases [7]. The proteolytic processing of APP C-terminal fragment C99 by γ-secretase proceeds through initial endoproteolysis of C99 and sequential trimming of long Aβ peptide intermediates along two main pathways: C99 → Aβ49 → Aβ46 → Aβ43 → Aβ40 and C99 → Aβ48 → Aβ45 → Aβ42 → Aβ38 [8] (Figure 1A). Many FAD mutations in PSEN1 or APP alter this processivity, skewing cleavage toward the production of longer, more aggregation-prone Aβ species such as Aβ42. The resulting elevations of the Aβ42/Aβ40 ratio, which is strongly associated with oligomeric Aβ assembly and early plaque deposition, has long been viewed as a central pathogenic trigger in AD [9].

However, growing evidence reveals that many PSEN1 FAD mutations do not conform to this conventional model. These mutations fail to elevate Aβ42 or the Aβ42/Aβ40 ratio, yet they still cause early-onset AD [10]. These and other findings challenge the Aβ-centric view of AD and have prompted investigation of alternative disease mechanisms. One such hypothesis, recently proposed by us, is the “stalled complex hypothesis”, which posits that FAD mutations compromise processive proteolysis by γ-secretase. Instead of driving Aβ aggregation, these mutations lead to accumulation of uncleaved or partially cleaved APP fragments bound to γ-secretase (i.e., stalled enzyme–substrate (E-S) complexes) that can trigger synaptic degeneration independently of Aβ production [11].

Our biochemical and structural investigations—involving mass spectrometry, cryo-electron microscopy, molecular dynamics simulations, and fluorescence lifetime imaging microscopy—demonstrated that twelve selected PSEN1 mutations stabilize the γ-secretase E-S complex and hinder its catalytic turnover [12,13]. In the model organism *C. elegans*, we further found that neuronal co-expression of FAD-mutant C99 and PSEN1 or expression of FAD-mutant PSEN1 alone can trigger synaptic loss and reduced lifespan in the absence of Aβ42 or near absence of Ab production, implicating stalled γ-secretase E-S complexes as pathogenic drivers [12,13]. These findings suggest that the key pathogenic trigger of FAD may not be aggregation-prone forms of Aβ; instead, prolonged substrate binding and impaired enzymatic release of proteolytic intermediates can induce a gain of neurotoxic function.

In this study, we investigated the effects of five FAD-associated PSEN1 mutations under investigation by the Dominantly Inherited Alzheimer Network (DIAN) [14] on γ-secretase activity. Using in vitro cleavage assays with purified enzyme and substrate, we observed that these FAD mutations significantly impaired initial endoproteolytic (ε) cleavage of APP C99 substrate and altered specific downstream carboxypeptidase trimming events. Complementary fluorescence lifetime imaging microscopy (FLIM) analysis in cultured cells co-expressing FAD-mutant PSEN1 and wild-type C99 substrate indicate that each of these FAD mutations stabilize γ-secretase E-S complexes. Collectively, these results reinforce the stalled complex hypothesis and support a critical pathogenic mechanism in FAD that functions independently of Aβ accumulation. Our findings support a revised model of AD pathogenesis in which disruption of proteolytic processing, rather than Aβ elevation alone, contributes to neuronal dysfunction and disease progression.

## 2. Materials and Methods

### 2.1. Construction of FAD-Mutant γ-Secretase Expression Vectors

FAD mutant constructs of γ-secretase were generated using ligation-independent cloning (LIC), as previously described [12,13]. Mutations were introduced into the monocistronic pMLINK-PS1 expression vector by site-directed mutagenesis using the QuikChange Lightning Multi Site-Directed Mutagenesis Kit (Agilent Technologies, Santa Clara, CA, USA). A previously constructed tricistronic pMLINK vector encoding Aph1, Nicastrin, and Pen-2 was used in combination with the monocistronic PS1 vector to generate tetracistronic expression constructs. Both vectors contained LINK1 and LINK2 sequences flanking the gene cassettes; LINK1 harbors a PacI restriction site, while LINK2 contains both PacI and SwaI sites. The mutant pMLINK-PS1 vector was digested with PacI, and the PS1 fragment was purified from a 1% agarose gel. Simultaneously, the tricistronic pMLINK-Aph1-Nicastrin-Pen-2 vector was linearized using SwaI, gel purified, and prepared for LIC. The PS1 insert and linearized backbone were treated with T4 DNA polymerase in the presence of dCTP and dGTP, respectively, at room temperature for 20 min. The treated PS1 fragment was then ligated into the linearized vector to generate tetracistronic pMLINK constructs encoding WT or mutant γ-secretase complexes.

### 2.2. Expression and Purification of γ-Secretase Complexes

Wild-type and FAD-mutant γ-secretase complexes were expressed in Expi293F cells (Thermo Fisher Scientific, Waltham, MA, USA) according to the manufacturer’s protocol. Cells were cultured in Expi293 expression medium until a density of 3 × 10^6^ cells/mL was reached. At this point, the medium was changed with fresh media, and the cells were transfected with 100 µg of the tetracistronic construct using Expifectamine 293 Transfection reagent in 6 mL of Opti-MEM. Following a 10-min incubation at room temperature, the transfection mixture was added to the cultured cells. The cells were harvested after 72 h by centrifugation. The cell pellets were resuspended in lysis buffer (50 mM MES, pH 6.0, 150 mM NaCl, 5 mM CaCl_2_, 5 mM MgCl_2_) and lysed using a French press (GlenMills, Clifton, NJ, USA). Unlysed cells were removed by centrifugation at 3000× *g* for 10 min, and membrane fractions were collected by ultracentrifugation at 100,000× *g* for 1 h. The membranes were solubilized in extraction buffer (50 mM HEPES, pH 7.0, 150 mM NaCl, 1% CHAPSO) and incubated on ice for 1 h. The solubilized fraction was clarified by ultracentrifugation and incubated overnight at 4 °C with anti-FLAG M2 agarose beads (Sigma-Aldrich, St. Louis, MO, USA) in TBS containing 0.1% digitonin. After washing, γ-secretase complexes were eluted with 0.2 mg/mL FLAG peptide in TBS containing 0.1% digitonin. Enzyme purity was verified by silver staining, and each component of γ-secretase was identified and verified by Western blot analysis. Purified complexes were aliquoted and stored at −80 °C until use.

### 2.3. In Vitro γ-Secretase Activity Assays

The enzymatic activity of purified γ-secretase complexes was measured as previously described [12,15]. Briefly, 30 nM of WT or mutant enzyme was preincubated for 30 min at 37 °C in assay buffer (50 mM HEPES, pH 7.0, 150 mM NaCl, 0.25% CHAPSO, 0.1% phosphatidylcholine, 0.025% phosphatidylethanolamine). The reactions were initiated by adding 5 µM C100-FLAG substrate [16] and incubated at 37 °C for 16 h. The reactions were quenched by flash freezing in liquid nitrogen and stored at –20 °C until analysis.

### 2.4. LC-MS/MS Analysis of Tri- and Tetrapeptide Products

Small peptide trimming products were analyzed by liquid chromatography–tandem mass spectrometry (LC-MS/MS) using an electrospray ionization quadrupole time-of-flight mass spectrometer (Q-TOF Premier, Waters Corp., Milford, MA, USA), as previously described [15]. The peptides were separated on a C18 reverse-phase column and eluted using a gradient of 0.08% aqueous formic acid, acetonitrile, isopropanol, and a 1:1 acetone/dioxane mixture. The three most abundant product ions were selected for collision-induced dissociation (CID) and MS/MS analysis. Synthetic peptide standards (>98% purity, New England Peptide) were analyzed in parallel at concentrations ranging from 32.5 nM to 1000 nM to generate standard curves. Quantification was performed by summing the intensities of the three most abundant ions within a 0.02 Da mass window. All data were collected in “V” mode.

### 2.5. Quantification of AICD Products by MALDI-TOF MS

AICD-FLAG coproducts were immunoprecipitated from γ-secretase reactions using anti-FLAG M2 agarose beads (Sigma-Aldrich) in buffer containing 10 mM MES (pH 6.5), 10 mM NaCl, and 0.05% n-dodecyl-β-D-maltoside (DDM). After 16-h incubation at 4 °C, the bound AICD was eluted with a 1:1 mixture of acetonitrile and water containing 0.1% trifluoroacetic acid. Synthetic AICD standards were prepared at concentrations ranging from 100 nM to 2000 nM for calibration. A fixed concentration of 50 nM ProteoMass Insulin (Sigma-Aldrich) was included in all samples as an internal standard. The samples were analyzed using a Autoflex MALDI-TOF mass spectrometer (Bruker Corp., Billerica, MA, USA) under standard operating conditions.

### 2.6. Quantification of Aβ40 and Aβ42 by ELISA

Quantification of Aβ40 and Aβ42 derived from in vitro cleavage assays with purified γ-secretase complexes and C100Flag substrate was performed using specific ELISA kits (Thermo Fisher Scientific, Waltham, MA, USA) following the manufacturer’s protocol.

## 3. Results

### 3.1. FAD PSEN1 Mutations Impair Proteolysis of C99 by γ-Secretase

To investigate the functional impact of FAD-associated mutations on γ-secretase activity, we generated five tetracistronic DNA constructs encoding mutant forms of the γ-secretase complex. The selected PSEN1 variants—C92S, M84V, Y115H, T116I, and M139V (Figure 1B)—correspond to mutations identified in the Dominantly Inherited Alzheimer Network (DIAN) registry [14]. The WT and mutant constructs were each transiently transfected into HEK293 cells, and the resulting γ-secretase complexes were purified for in vitro analysis. Purified WT and mutant proteases (30 nM; Appendix A) were incubated with saturating concentrations of the APP C99-based recombinant substrate C100-FLAG (5 µM) at 37 °C for 16 h. Due to the analytical challenges posed by the size and hydrophobicity of Aβ peptides, we focused our analysis on quantifying proteolytic coproducts (Figure 1A), using these values to indirectly calculate Aβ product concentrations as described previously [15]. Moreover, these co-products are terminal products that—unlike Aβ intermediates—are not subject to further degradation, allowing quantitation of the effects of each FAD mutation on each cleavage step.

To assess the efficiency of the initial ε-cleavage event, we quantified the production of the FLAG-tagged APP intracellular domain (AICD-FLAG; Figure 1A) using MALDI-TOF mass spectrometry. Synthetic FLAG-tagged AICD50–99 and AICD49–99 peptides were used to generate standard curves (Figure 1C), allowing determination of concentrations of AICD species corresponding to cleavage at the Aβ49 and Aβ48 sites (Appendix A). Insulin was included as an internal standard across all experimental and control samples. Quantitative analysis of the AICD peptides revealed that all five PSEN1 mutants exhibited substantially reduced production of AICD50–99 and AICD49–99 compared to WT (Figure 1D), indicating impaired ε-cleavage efficiency. Quantification of the generation of AICD50–99 and AICD59–99 also measures formation of coproducts Aβ49 and Aβ48, respectively. These findings are consistent with the previous reports of diminished endoproteolysis in all but one of twelve FAD-associated PSEN1 mutations analyzed by this method [12,13].

We next evaluated the processive cleavage activity of γ-secretase along the two canonical pathways: Aβ49 → Aβ46 → Aβ43 → Aβ40 and Aβ48 → Aβ45 → Aβ42 → Aβ38. Using LC-MS/MS, we quantified tripeptide and tetrapeptide coproducts released during each cleavage step, (Figure 2A,B). Standard curves were generated from synthetic ITL, VIV, IAT, VIT, TVI, and VVIA peptides, enabling accurate quantification across all enzyme variants. Our analysis revealed that all mutant proteases displayed deficiencies in one or more processive cleavage steps. Quantitative analysis of AICD and small-peptide coproducts by LC-MS/MS and MALDI-TOF enabled determination of the percent cleavage achieved for each carboxypeptidase trimming event during processive γ-secretase proteolysis of the APP C99 substrate (Figure 2C,D). This analysis revealed that mutant enzymes M84V and Y115H exhibited deficiencies in the Aβ49 → Aβ46, Aβ48 → Aβ45, Aβ45 → Aβ42 and Aβ42 → Aβ38 cleavage steps compared to the WT enzyme. Similarly, C92S mutant is deficient in Aβ46 → Aβ43 and Aβ42 → Aβ38 cleavage steps. The T116I and M139V mutant were deficient in Aβ45 → Aβ42 and Aβ42 → Aβ38 cleavage step compared to the WT enzyme. By analyzing the extent of degradation and production of each Aβ peptide, we were able to calculate the final levels of each Aβ peptide in the enzyme reaction mixtures (Table 1).

To validate the calculated concentrations of Aβ peptides obtained by LC-MS/MS, we performed ELISAs specific for Aβ40 and Aβ42 peptides (Figure 3A). The ELISA results closely aligned with the LC-MS/MS-derived peptide concentrations (Figure 3B) and confirmed that all five PSEN1 mutants produced lower levels of both Aβ40 and Aβ42 compared to WT. Mutants Y115H and T116I showed an increased Aβ42/Aβ40 ratio by both ELISA and LC-MS/MS (Figure 3). In contrast, Aβ42/Aβ40 produced from the C92S mutant enzyme was equivalent to that of WT by both methods. For, M84V and M139V mutant enzymes, the Aβ42/Aβ40 ratio is slightly increased compared with that of WT when measured by ELISA but was statistically similar to that of WT when calculated by LC-MS/MS.

The ELISA results confirmed the final Aβ40 and Aβ42 concentrations from all enzyme reactions as calculated by LC-MS/MS and thereby provided confidence in the calculations of the final concentrations of other Aβ peptides. However, note that the calculated concentrations of Aβ43 produced from the PSEN1 Y115H mutant enzyme and of Aβ48 from the PSEN1 T116I mutant enzyme were negative, which is not possible. Calculated Aβ43 from PSEN1 Y115H protease was only slightly negative, consistent with complete conversion of Aβ48 to Aβ45. However, calculated Aβ48 from PSEN1 T116H protease was substantially negative (−46.8 nM). The reason for this is unclear but may reflect either under-measurement of AICD50–99 co-product of Aβ48 production from C100-FLAG or over-measurement of VIT coproduct of Aβ45 production from Aβ48 by this mutant enzyme.

### 3.2. FAD Mutations Stabilize the γ-Secretase/Substrate Interaction

To assess the impact of FAD-associated PSEN1 variants on γ-secretase enzyme–substrate (E-S) complex stability, we performed fluorescence lifetime imaging microscopy (FLIM) in intact HEK293 cells in which endogenous PSEN1 and PSEN2 were knocked out via CRISPR/Cas9-mediated gene editing [17]. Cells were co-transfected with a C99-miRFP720 construct—comprising human APP C99 with an N-terminal signal peptide and C-terminal near-infrared miRFP720—and either wild-type (WT) or one of the five FAD PSEN1 mutants. After fixation and permeabilization, samples were stained with mouse α-6E10 (recognizing the C99/Aβ N-terminus) and rabbit α-nicastrin (proximal to bound substrate), followed by Alexa Fluor 488-conjugated anti-mouse and Cy3-conjugated anti-rabbit secondary antibodies. Energy transfer from Alexa 488 to Cy3, detected as a shortened Alexa 488 fluorescence lifetime, reports E-S complex formation (Figure 4A). Each PSEN1 mutant exhibited a significant decrease in donor lifetime compared to WT (Figure 4B), reflecting stabilization of γ-secretase E-S complexes.

## 4. Discussion

In this study, we demonstrate that five FAD–associated PSEN1 mutations (C92S, M84V, Y115H, T116I, and M139V) markedly impair the processive proteolysis of the APP C99 substrate by γ-secretase. Using in vitro assays with purified proteins, we observed significant reductions in ε-cleavage efficiency, as evidenced by decreased AICD50–99 and AICD49–99 coproducts across all mutants. Subsequent trimming steps along both the Aβ49 → Aβ40 and Aβ48 → Aβ38 pathways were also compromised, with mutant complexes exhibiting deficiencies in one or more carboxypeptidase events. Complementary ELISA and LC-MS/MS quantification confirmed that each PSEN1 variant produced lower overall levels of both Aβ40 ad Aβ42 compared to wild-type (WT) enzyme, with one mutant (C92S) showing statistically equal Aβ42/Aβ40 compared with WT. Finally, FLIM analysis in PSEN1/2-knockout HEK293 cells revealed all five FAD mutations significantly stabilized γ-secretase-substrate (E-S) complexes, supporting a stalled-process model of enzymatic dysfunction.

These findings align with and extend the “stalled complex” hypothesis, which posits that FAD mutations cause a toxic gain of function via prolonged retention of substrate on the protease, rather than solely by shifting the Aβ42/Aβ40 ratio [12]. Our results are consistent with previous reports demonstrating that many PSEN1 mutations reduce processivity without necessarily raising the Aβ42/Aβ40 ratio [10,15] and reinforce the concept that stabilized E-S complexes can trigger synaptic degeneration independently of Aβ production [12].

From a therapeutic standpoint, our data argue against γ-secretase inhibition as a viable strategy for FAD or sporadic AD, since further reducing proteolytic activity may exacerbate the accumulation of stalled complexes along with on-target reduction of Notch signaling. Indeed, γ-secretase inhibitors caused cognitive worsening in advanced clinical trials [18,19]. Instead of inhibitors, compounds that enhance processivity—so-called γ-secretase modulators—offer a promising alternative by promoting the trimming of Aβ intermediates and release of products, thereby reducing the lifetime of E-S complexes. Such agents could restore deficient trimming activity across multiple cleavage steps without altering global substrate specificity, potentially ameliorating synaptic dysfunction driven by stalled complexes.

Several limitations temper our conclusions and warrant future investigation. All assays were conducted with purified enzyme and substrate in detergent-solubilized systems; although the prior work suggests concordance with lipid-reconstituted proteoliposomes [15], the native membrane environment may modulate processivity and complex stability. Additionally, while FLIM provides a robust measure of E-S complex stabilization in cells, it remains to be seen whether similar effects occur in neurons or in vivo. Finally, the relevance of stalled γ-secretase complexes to sporadic AD—with its heterogeneous genetic and environmental risk factors—requires exploration. Addressing these questions in physiologically relevant models will be critical to validating the stalled complex hypothesis and guiding the development of targeted γ-secretase activators for AD.

## 5. Conclusions

Our findings provide compelling evidence that FAD mutations in PSEN1 disrupt γ-secretase function by impairing both ε-cleavage and subsequent processive trimming of the APP C99 substrate. The five FAD-associated mutations examined here—C92S, M84V, Y115H, T116I, and M139V—not only led to diminished production of AICD and Aβ species but also stabilized enzyme–substrate complexes, as demonstrated by FLIM imaging in intact cells. These results support the stalled complex hypothesis and suggest that the pathogenic mechanism of FAD may involve toxic retention of APP intermediates within γ-secretase, rather than simply increased Aβ42/Aβ40 ratios with resultant aggregation [11].

Taken together, this study highlights a critical shift in our understanding of γ-secretase dysfunction in AD, moving beyond Aβ42 as the pathogenic trigger to a broader evaluation of catalytic processivity and enzyme–substrate dynamics. By uncovering the proteolytic and structural consequences of specific PSEN1 mutations, we lay a foundation for the development of therapeutic strategies aimed at restoring γ-secretase processivity and preventing the formation of pathogenic stalled complexes. For instance, rescuing deficient γ-secretase proteolysis might be achieved by allosteric modulation. Future studies using neuronal models and in vivo systems will be essential to translate these mechanistic insights into potential clinical interventions for familial and possibly sporadic forms of AD.

## Figures and Tables

**Figure 1 biomolecules-15-00955-f001:**
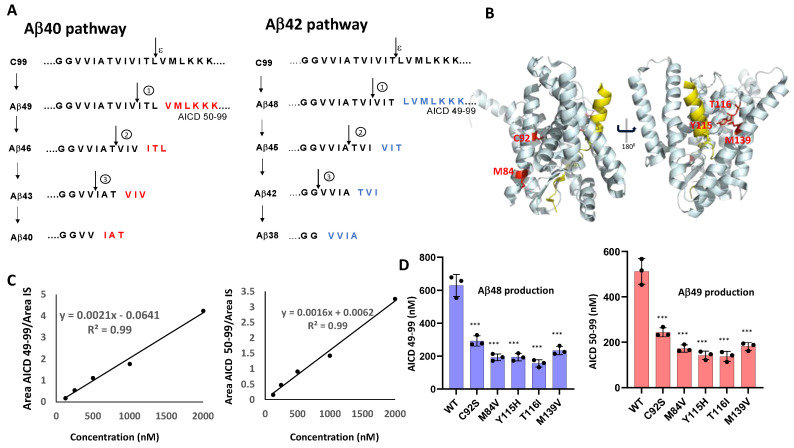
Effects of FAD-Associated PSEN1 Mutations on ε-Cleavage of C99 by γ-Secretase. (**A**) Schematic representation of γ-secretase-mediated proteolysis of APP C99 substrate, highlighting sequential cleavage events. (**B**) Cryo-EM-based model of APP C99 (yellow) bound to the PSEN1-containing γ-secretase complex (other complex components nicastrin, Aph-1, and Pen-2 omitted for clarity), with FAD-associated PSEN1 mutations examined in this study shown in red. (**C**) Standard curves for AICD50–99 and AICD49–99 (coproducts of Aβ49 and Aβ48 generation, respectively) were generated using synthetic peptides and MALDI-TOF MS, with 50 nM insulin as an internal standard (IS). (**D**) Quantification of AICD products from WT and mutant γ-secretase reactions, based on MALDI-TOF MS measurements using the established standard curves. Data represent mean values from three independent experiments (n = 3). Statistical significance was determined using unpaired two-tailed *t*-tests; *** *p* < 0.001. Throughout the manuscript, purple bars denote quantification of cleavages along the Aβ48 → Aβ38 pathway, while red bars denote quantification of cleavages along the Aβ49 → Aβ40 pathway.

**Figure 2 biomolecules-15-00955-f002:**
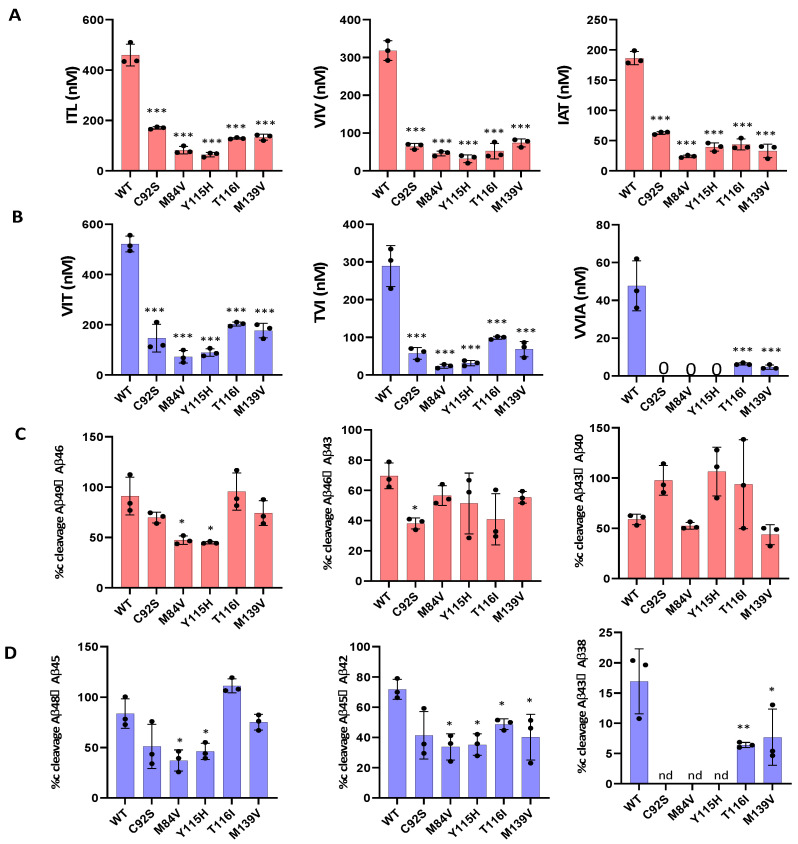
FAD-mutant PSEN-1 Affects Processive Proteolysis of C99 by γ-Secretase. (**A**) Bar graphs illustrating all coproduct formation for specific mutations for the Aβ40 pathway. (**B**) Bar graphs illustrating all coproduct formation for specific mutations for the Aβ42 pathway. (**C**) Percentage cleavage efficiency was calculated for Aβ40 carboxypeptidase (trimming) pathway. (**D**) Percentage cleavage efficiency was calculated for Aβ42 carboxypeptidase (trimming) pathway. Steps yielding no detectable coproduct (“nd”) are indicated. Each graph represents n = 3, and statistical significance was determined using unpaired two-tailed *t*-tests comparing FAD mutants with WT (* *p* < 0.05, ** *p* < 0.01, *** *p* < 0.001).

**Figure 3 biomolecules-15-00955-f003:**
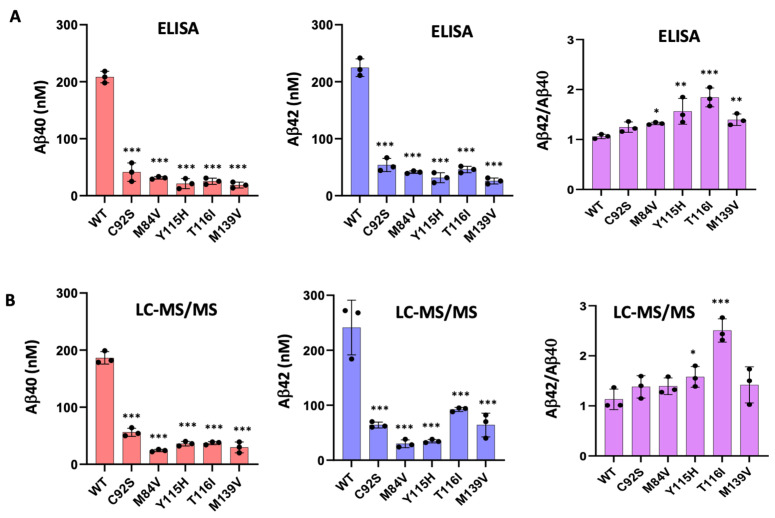
Validation of Aβ40 and Aβ42 Quantification by LC-MS/MS Using ELISA. (**A**) Aβ40 and Aβ42 levels were quantified in reaction mixtures containing purified γ-secretase complexes incubated with C100Flag substrate, using specific ELISAs. The resulting Aβ42/Aβ40 ratios are shown alongside absolute peptide concentrations. (**B**) Independent calculation of Aβ40 and Aβ42 quantification and Aβ42/Aβ40 ratios were performed from LC-MS/MS analysis of proteolytic coproducts. Data represent mean values from three independent experiments (n = 3). Statistical significance was determined using unpaired two-tailed *t*-tests comparing FAD-mutant enzymes to wild-type (WT) γ-secretase. * *p* ≤ 0.05; ** *p* ≤ 0.01; *** *p* ≤ 0.001.

**Figure 4 biomolecules-15-00955-f004:**
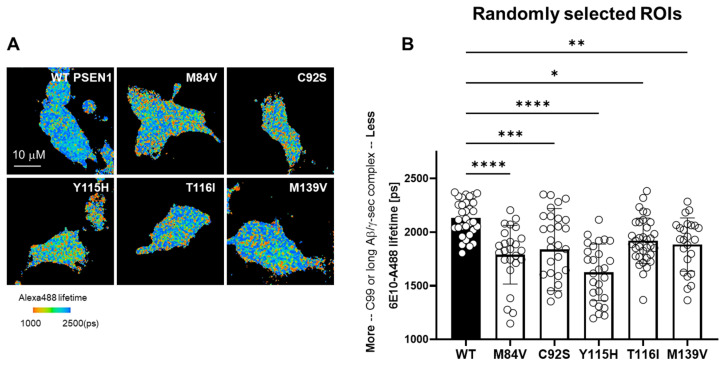
Stabilization of γ-Secretase Enzyme–Substrate Complexes by FAD-Associated PSEN1 Mutations. (**A**) PSEN1/2 double knockout (dKO) HEK293 cells were co-transfected with C99–720 and either wild-type (WT) or FAD-mutant PSEN1 constructs. Following transfection, cells were immunostained with primary antibodies targeting the N-terminus of C99/Aβ (mouse monoclonal 6E10) and nicastrin (rabbit polyclonal NBP2-57365), followed by secondary antibodies conjugated to Alexa Fluor™ 488 (donor) and Cy3 (acceptor), respectively. Fluorescence lifetime imaging microscopy (FLIM) was performed to measure the donor (6E10-Alexa Fluor™ 488) fluorescence lifetime. A decrease in donor lifetime indicates energy transfer to the acceptor and reflects stabilized enzyme–substrate (E-S) interactions. Scale bar = 10 µm. (**B**) Quantitative analysis of 6E10-Alexa Fluor™ 488 lifetimes from randomly selected regions of interest (ROIs; N = 40–47 ROIs from 6–8 cells) revealed significant stabilization of γ-secretase E-S complexes in cells expressing FAD-mutant PSEN1 compared to WT controls. Statistical analysis was performed using one-way ANOVA followed by Tukey’s multiple comparisons test. (*p* > 0.05); * *p* < 0.05; ** *p* < 0.01; *** *p* < 0.001; **** *p* < 0.0001.

**Table 1 biomolecules-15-00955-t001:** Calculated concentration (nM) of all Aβs produced from proteolysis of APP substrate by wild-type and FAD mutant γ-secretase.

	Aβ49	Aβ46	Aβ43	Aβ40	Aβ48	Aβ45	Aβ42	Aβ38
**WT**	52.5	142	131.5	186.4	108.2	132.3	241.3	47.6
**C92S**	74.3	105.1	8.8	56.1	145.5	82.6	64.5	nd
**M84V**	90.2	36.3	22.1	24.4	120.7	42.3	30.1	nd
**Y115H**	78.4	32.7	−4.3	36.7	104.7	54.6	34.6	nd
**T116I**	8.7	76.3	15.3	37.4	−46.8	104.1	92.3	6.3
**M139V**	48.1	59.3	44.3	29.6	57.6	108.2	64.6	4.6

## Data Availability

The original contributions presented in this study are included in the article/Appendix A. The raw data supporting the conclusions of this article will be made available by the authors on request.

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
