# Peer review of "Presenilin-1 Familial Alzheimer Mutations Impair γ-Secretase Cleavage of APP Through Stabilized Enzyme–Substrate Complex Formation"

_biomolecules, 2025, doi:10.3390/biom15070955_

Round 1

Reviewer 1 Report

Comments and Suggestions for Authors

Abstract:

line 15: Abeta is a peptide, not a protein; If mentioned in here, "stalled complex" must be explained; it is not a commonly accepted term in the field; "five additional...": mention these here; it is unclear why DIAN is mentioned in here, why is this important?..."biochemical analysis...." this sentence is difficult to read, rephrase; 

Intro: ...10% of indivudials over age 65....: rephrase, over 65 can be 100; ...two main pathways...: repetitive to abstract; line 72: it is not either or...it is both 1) effect on proteolysis 2) effect on aggregation through increased Ab42 production; claim to absolute right is scientifically not correct;

Results: Table 1: giving of negative concentration is non-sense...see Ab43 and 48 concentrations in the table; quantification in absolute concentrations by MALDI is questioned; compare with literature values;

Supplemental material: WB data should indicate KDa of bands shown; MW markers must be shown.

Author Response

Abstract:

line 15: Abeta is a peptide, not a protein.

Response: This has been corrected.

 If mentioned in here, "stalled complex" must be explained. It is not a commonly accepted term in the field.

Response: “Stalled complex” was already defined in the abstract: “FAD mutations lead to stalled γ-secretase/substrate complexes that trigger synaptic loss independently of Aβ production.” See lines 19-20).

"five additional...": mention these here.

Response: The specific mutations are now mentioned in the abstract.

It is unclear why DIAN is mentioned in here, why is this important?

Response: This has been removed

..."biochemical analysis...." this sentence is difficult to read, rephrase (

Response: The sentence seems clear as it is; however, we now add the word “subsequent” in a way that should be helpful: “Comprehensive biochemical analysis revealed that all five mutations led to substantially reduced initial proteolysis of C99 to Aβ49 or Aβ48 as well as deficiencies in one or more subsequent trimming steps.

Intro:

...10% of individuals over age 65....: rephrase, over 65 can be 100

Response: We have replaced this phrase with the following: “…affecting 5% of people aged 65 to 74, 13% of people aged 75-84, and 33% of people aged 85 and older”. We also replaced the original Ref. 1 with a more appropriate reference.

...two main pathways...: repetitive to abstract.

Response: It is not repetitive; it expands on what is succinctly stated in the abstract. We believe this is critical to understanding by the reader.

line 72: it is not either or...it is both 1) effect on proteolysis 2) effect on aggregation through increased Aβ42 production. Claim to absolute right is scientifically not correct.

Response: We have rephrased parts of this sentence to make it clear that we observed synaptic loss in the absence of Aβ42 or total Aβ production in our previous study (Devkota et al, Cell Reports, 2024). This evidence suggests that stalled γ-secretase enzyme-substrate complexes, not Aβ, trigger the synaptic degeneration. We are not claiming that this is absolutely correct: We use the terms “implicating” and “suggest”.)

Results: Table 1: giving of negative concentration is non-sense...see Ab43 and 48 concentrations in the table

Response: We completely agree and have added a new paragraph (lines 239-249):

“The ELISA results confirmed the final Aβ40 and Aβ42 concentrations from all enzyme reactions as calculated by LC-MS/MS and thereby provided confidence in the calculations of the final concentrations of other Aβ peptides. However, note that the calculated concentrations of Aβ43 produced from the PSEN1 Y115H mutant enzyme and of Aβ48 from the PSEN1 T116I mutant enzyme were negative, which is not possible. Calculated Aβ43 from PSEN1 Y115H protease was only slightly negative, consistent with complete conversion of Aβ48 to Aβ45. However, calculated Aβ48 from PSEN1 T116H protease was substantially negative (-46.8 nM). The reason for this is unclear but may reflect either under-measurement of AICD50-99 co-product of Aβ48 production from C100-FLAG or over-measurement of VIT coproduct of Aβ45 production from Aβ48 by this mutant enzyme.”

quantification in absolute concentrations by MALDI is questioned; compare with literature values

Response: We generated standard curves with synthetic AICD50-99-FLAG and AICD49-99-FLAG using MALDI-MS, as shown in Fig. 1C. The quantified concentrations of these coproducts generated from each enzyme reaction fell within the range of the standard curves and are therefore valid. Moreover, we have recently reported this MALDI-MS method for quantifying  AICD-FLAG coproducts (Arafi et al, eLife, 2025). The results shown in that report for final AICD-FLAG coproduct levels after incubation with WT enzyme are closely similar to what we show in the present manuscript. Furthermore, these results are consistent with what we previously reported for AICD-FLAG production from WT enzyme using a different method, a combination of (1) quantitative western blotting of total AICD-FLAG production and (2) determination of the ratio of AICD50-99-FLAG to AICD49-99 as calculated by MALDI-MS (Devkota et al., J Biol Chem, 2021).

Supplemental material: WB data should indicate KDa of bands shown; MW markers must be shown.

We now provide the Western blot data with MW markers and KDa of bands indicated.

Reviewer 2 Report

Comments and Suggestions for Authors

This elegant study shows that five mutations in PSEN1 substantially reduced initial proteolysis of C99. It is a useful contribution to the field but does have several shortcomings:

  1. The title needs to be improved, eg: ‘Presenilin-1 familial Alzheimer’s mutations impair γ-secretase cleavage of APP through stabilized enzyme-substrate complex formation’
  2. The Introduction describes how LOAD affects 10% of the population. As the focus is mutations on PSEN1 it would be useful to know how many people are affected or the relative contributions of mutations in APP, PSEN1 and PSEN2 to EOAD.
  3. ‘Alzheimer’s disease’ is abbreviated to ‘AD’ on line 33 but. This abbreviation needs to be used consistently throughout the manuscript.
  4. A synapse is the specialized junction between two neurons. The authors repeatedly refer to ‘synaptic loss’ Does this mean the loss of both pre- and post-synaptic neurons? ‘Neuron loss’ would be much clearer.
  5. The text in Figure 1 is too small to read. This needs to be corrected.
  6. The authors need to consider the possibility of allosteric modulators of γ-secretase providing a starting point for the discovery of potential disease-modifying new medicines for AD.

Author Response

The title needs to be improved, eg: ‘Presenilin-1 familial Alzheimer’s mutations impair γ-secretase cleavage of APP through stabilized enzyme-substrate complex formation’

Response: We have replaced the original title with the suggested one.

The Introduction describes how LOAD affects 10% of the population. As the focus is mutations on PSEN1 it would be useful to know how many people are affected or the relative contributions of mutations in APP, PSEN1 and PSEN2 to EOAD.

Response: We have modified the sentence that introduces FAD accordingly: “While most cases of AD are sporadic and late-onset, a small subset (1-5%) suffer from familial Alzheimer’s disease (FAD), which is dominantly inherited and typically manifests in midlife.” Note that the range is broad, because the percentages of FAD reported in the literature are inconsistent.

‘Alzheimer’s disease’ is abbreviated to ‘AD’ on line 33 but. This abbreviation needs to be used consistently throughout the manuscript.

Response: Corrected.

A synapse is the specialized junction between two neurons. The authors repeatedly refer to ‘synaptic loss’ Does this mean the loss of both pre- and post-synaptic neurons? ‘Neuron loss’ would be much clearer.

Response: In our cited Cell Reports paper, we used transgenic C. elegans lines expressing WT or FAD-mutant PSEN1 + C99 in neurons. The parental line expressed synaptobrevin fused with GFP in GABAergic neurons, allowing visualization of presynaptic termini along the dorsal and ventral nerve cords by confocal fluorescence microscopy. Age-dependent loss of presynaptic termini was observed with FAD mutations. The neuronal cell bodies, however, could still be seen. Thus, “synaptic loss” is the appropriate term for what we observed, not “neuronal loss”.

The text in Figure 1 is too small to read. This needs to be corrected.

Response: Corrected.

The authors need to consider the possibility of allosteric modulators of γ-secretase providing a starting point for the discovery of potential disease-modifying new medicines for AD.

Response: We have added a new sentence to the final paragraph of the Conclusions section, immediately after mentioning that our findings have therapeutic implications: “For instance, rescuing deficient γ-secretase proteolysis might be achieved by allosteric modulation.” (lines 345-346).

Round 2

Reviewer 1 Report

Comments and Suggestions for Authors

All points previously raised have sufficiently been addressed.